# Sociodemographic and Work-Related Variables Affecting Knowledge of, Attitudes toward, and Skills in EBNP of Nurses According to an EBPPQ

**DOI:** 10.3390/ijerph19148548

**Published:** 2022-07-13

**Authors:** Katarzyna Młynarska, Elżbieta Grochans, Magdalena Sylwia Kamińska, Anna Maria Cybulska, Mariusz Panczyk, Ewa Kupcewicz

**Affiliations:** 1Department of Nursing, Collegium Medicum University of Warmia and Mazury in Olsztyn, 14C Żołnierska St., 10-719 Olsztyn, Poland; katarzyna.mlynarska@uwm.edu.pl (K.M.); ekupcewicz@wp.pl (E.K.); 2Department of Nursing, Faculty of Health Sciences, Pomeranian Medical University in Szczecin, 48 Żołnierska St., 71-210 Szczecin, Poland; elzbieta.grochans@pum.edu.pl (E.G.); anna.cybulska@pum.edu.pl (A.M.C.); 3Subdepartment of Long-Term Care, Department of Social Medicine, Faculty of Health Sciences, Pomeranian Medical University in Szczecin, 48 Żołnierska St., 71-210 Szczecin, Poland; 4Department of Education and Research in Health Sciences, Faculty of Health Sciences, Medical University of Warsaw, 14/16 Litewska St., 00-518 Warszawa, Poland; mariusz.panczyk@wum.edu.pl

**Keywords:** evidence-based nursing practice, Evidence-Based Practice Profile Questionnaire, sociodemographic variables

## Abstract

(1) This study examines sociodemographic and work-related variables to determine their impact on the knowledge of, attitudes toward, and skills in Evidence-Based Nursing Practice (EBNP). (2) The study included 830 nurses from four voivodships in Poland, Dolnośląskie, Łódzkie, Podlaskie, and Zachodniopomorskie and was conducted by the diagnostic survey method, using the questionnaire technique. The following research tools were applied in the study: an Evidence-Based Practice Profile Questionnaire (EBPPQ) and a survey questionnaire developed by the authors, containing questions on sociodemographic data and work-related variables. (3) Sociodemographic variables with an impact on the knowledge of, attitudes toward, and skills in EBNP include age, marital status, and educational background. Older nurses find it problematic to apply EBNP, and their level of relevant knowledge and skills is lower; whereas, those with university education possess the best EBNP-related skills, and they are also more eager to expand them and apply them in their work. The work-related variables with an impact on the knowledge of, attitudes toward, and skills in EBNP among nurses include work experience, which has a negative effect on applying EBNP and the skills associated with it. The type of school from which the nurses graduated and the nursing specialisation training also have a strong impact on expanding nurses’ competence in EBNP.

## 1. Introduction

Evidence-based medicine (EBM) is the paradigm of contemporary medicine, and it consists of applying credible scientific evidence concerning the safety and efficacy of diverse therapeutic medical interventions in patient clinical management [1,2]. It denotes the “integration of individual skills and knowledge with the best available, external clinical evidence from methodical studies” [3]. The aim of the EBM is to “consciously, unambiguously and reasonably apply the best and up-to-date scientific evidence when making decisions concerning the care of an individual patient” [2]. EBM is based on randomized controlled trials (RCT) and observational studies.

The term “evidence-based nursing practice” (EBNP) is used in reference to nursing activities. “Evidence-based nursing practice is oriented towards strengthening nurses’ professional position and promoting high-quality patient care” [4]. According to the International Council of Nurses (ICN), EBNP is defined as “an approach to solving problems in making clinical decisions which involves seeking the best and the latest evidence, clinical expertise and evaluations, and the value of patient’s care-related preferences” [5]. According to the Canadian Nurses Association (CNA) “evidence-based practice involves making decisions and is used to optimise patient management outcomes, to improve clinical practice and to ensure accountability in nursing” [6]. According to CNA, best practice guidelines are based on the strictest studies available, such as systematic reviews and randomised clinical trials, but they also include studies that are “well-established in expert opinions and consensus” [6].

Modern professional nursing is focused on evidence-based practice (EBP) or evidence-based nursing (EBN). Evidence-based practice aims at applying the best methods possible in the nursing process, which are justified by the findings of studies conducted by professionals [3].

EBNP is founded on the systematic and consistent use of the latest and most reliable scientific research findings in everyday nursing practice and eliminating costly unjustified interventions of low effectiveness [3,7].

The EBNP paradigm is considered in three mutually complementary dimensions: personal, organisational, and educational. The EBNP personal dimension includes: self-reflection, critical thinking, and permanent education and skill development. The organisational dimension is characterised by the clinical management model assumptions as a global quality assurance process, which guarantees the maintenance and permanent improvement of standards of care and developing public healthcare responsibility. Permanent education at the workplace according to nursing education is an education-related attribute. “It is the permanent, common, multi-level, organised and methodical improvement of knowledge and translation into practical skills of the ward staff at the workplace” [3].

The use of the latest specialist scientific periodicals in nursing contributes largely to the improvement in nurses’ knowledge and skills, inspires them to develop their scientific careers, and to permanent education. The International Council of Nurses set the priority areas for nursing activities in accordance with the strategic plan adopted for the years 2019–2023 during the 2019 ICN Congress in Singapore, which covered the following domains: development of the nurses’ role in assuring accessibility and quality of healthcare services; professionalism and promotion of the nursing profession; socioeconomic conditions of nursing; and healthcare system development [8]. The trends for nursing scientific research in Europe are set by the European Nursing Research Foundation (ENRF), founded by the European Federation of Nurses Associations (EFN) [8].

Although the tradition of nursing scientific activity in Poland dates back to the inter-war period, the first scientific studies in nursing were not published until the 1960s. The current level of nurse education was significantly affected by the “European Agreement of Nurses’ Training” of 1967. Starting in the mid-1990s, the transformation process in nurse training was associated with Poland’s accession to the European Union. Beginning in 2000, professional nurse training in the majority of Polish universities was fully adapted to the European standards, and the education is now at the university level [9].

Both undergraduate and postgraduate education are regulated by the relevant legislation. However, it was only when the Regulation of the Minister of Science and Higher Education on the standards of education was published, 12 July 2007 [10], that an up-to-date standard of education for the first and second cycle of nursing studies was introduced, taking into account the “Research in nursing” subject in the first cycle of study. Meanwhile, the 9 May 2012 Regulation of the Minister of Science and Higher Education on the standards of education for the fields of study of medicine, medicine–dentistry, pharmacy, nursing, and midwifery [11] provided an update on the standard of education for the nursing field of study. This Regulation placed the subject “Scientific research in nursing” in a group of sciences related to basic nursing care in the first cycle of study. In the second cycle of study, the standard included the subject “Scientific research in nursing” in a group of selected issues within the social sciences. However, only after completing the second cycle of study was an undergraduate student for the first time able to “define the principles of evidence-based medicine (EBM) and evidence-based nursing practice (EBNP)” and “knew the rules of preparing publications for nursing scientific periodicals and demonstrated knowledge of applying EBNP in one’s own professional practice or as part of a managed team” [11].

Considering the essence of research in nursing at every stage of developing professional qualifications, the “Scientific research in nursing” module was included in the 2003 framework syllabi in the general vocational training block for nurses [12]. New framework programmes of specialisation training for nurses have been in use since 2017 [13].

This study examines sociodemographic and work-related variables to determine their impact on the knowledge of, attitudes toward, and skills in EBNP of nurses according to the Evidence-Based Practice Profile Questionnaire (EBPPQ).

## 2. Materials and Methods

### 2.1. Settings and Design

The research was conducted from November 2018 to June 2019 among nurses working in various regions of Poland (Dolnośląskie, Łódzkie, Podlaskie, and Zachodniopomorskie voivodeships).

The criterion for inclusion in the study was age > 18 years, a valid license to practise a profession, at least one year of work experience in nursing, and informed consent to participate in the study. The exclusion criteria were: no consent to participate in the study, age < 18, an invalid license to practise a profession, and work experience < 1 year. The study was carried out using the random selection method.

The study was conducted in accordance with the guidelines of the Helsinki Declaration and was approved by the Ethics Committee of the Pomeranian Medical University in Szczecin (resolution no. KB-0012/256/06/18). The respondents were informed about the purpose of the research, about the possibility of resignation and withdrawal of consent at each stage of the study, and had the opportunity to ask questions and obtain comprehensive explanations. After agreeing to participate in the study, the respondents received a set of paper questionnaires.

In total, 1000 sets of questionnaires were distributed. The respondents could withdraw from the study at any time, without giving a reason. The respondents did not receive any remuneration for participating in the research. The completion of the questionnaires took about 15 min. After collecting the data and eliminating the incorrectly completed questionnaires, 830 packages (i.e., 83% of the total) were qualified for further statistical analysis.

### 2.2. Participants 

The study included 830 nurses from four voivodships in Poland: Dolnośląskie (19.2%), Łódzkie (17.3%), Podlaskie (33.7%) and Zachodniopomorskie (29.8%). Women accounted for a vast majority of the participants (95.2%); they were mostly 41–50 years old (41.9%), lived in a city with a population exceeding 100 thousand (35.4%), were in a formal relationship (64.3%), and were university graduates (43.9%) (Table 1). 

### 2.3. Research Instruments

The study was conducted by the diagnostic survey method, using the questionnaire technique. The following research tools were applied in the study:An Evidence-Based Practice Profile Questionnaire (EBPPQ) developed by a team of authors from the School of Health Sciences, University of South Australia: Maureen P. McEvoy, Marie T. Williams, and Timothy S. Olds, intended for examining nurses’ knowledge of, attitudes toward, and skills in EBP. The psychometric properties (in terms of reliability and accuracy) of the Polish language version of the EBPPQ were confirmed by a team at the Medical University of Warsaw, and it can be used for scientific purposes in groups of nursing students and practitioners [14]. All of the EBPPQ domains in the Polish language version are highly reliable, as the Cronbach alpha ranged from 0.800 to 0.972 [14].A survey questionnaire developed by the authors, containing questions on sociodemographic data (sex, age, place of residence, family and marital status, and educational background) and work-related variables (work experience, place of work, and work organisation system).

### 2.4. Statistical Analysis

Metric and non-metric variables were characterised by descriptive statistic parameters. The following were determined for variables expressed on a metric (quantitative) scale: measures of central tendency (M—mean), measures of variability (SD—standard deviation, CV—coefficient of variation). The measures of structure, sample size (N) and purity (%), were determined for the variables expressed on a non-metric scale.

The classic statistics based on testing zero hypotheses were used for statistical inference. The choice of statistical tests was affected by the type of the dependent variable measurement scale and the number of factor levels (number of independent variable variants). Testing zero hypotheses was complemented by determining the measures of effect (with 95% confidence of intervals) matched to the type of the statistical test used. The mathematical statistics methods concerning inter-sample (inter-group) differences included: Student’s t-test for unrelated samples with an estimation of Cohen’s d coefficient; a one-way analysis of variance (ANOVA) with an estimation of the η^2^ coefficient, and Scheffe’s post hoc test.

The statistical analysis methods used to assess the relationships between two or more variables involved the Pearson linear correlation coefficient (r) defining the linear correlation between two metric variables.

The default level of statistical significance—0.05—was adopted for all analyses. The calculations were performed with STATISTICA v. 13.3 (TIBCO Software Inc., Palo Alto, CA, USA).

A regression model (generalized linear model (GLM)) was used to explain the perception of, knowledge of, and skills in EBNP. Two regression models were analysed: the full model (all predictors) and the stepwise model (the stepwise model that best explains the variance of the dependent variable). All the independent variables (predictors) were entered into the full regression model simultaneously. Then, predictors were eliminated backwards from the full regression model in order to obtain the model that best explained the variance of the dependent variable. We tested the assumption of the noncollinearity of the predictors by calculating the tolerance and the variance inflation factor (VIF). The standardized regression coefficient (β) with 95% confidence interval (95% CI) was estimated for each predictor. The adjusted coefficient of determination (adjusted R-squared) was calculated to define the percentage of the explained variance of the dependent variable by the independent variables.

## 3. Results

### 3.1. Educational and Work-Related Profile of the Study Group

The majority of the 830 nurses graduated from a medical university (48.2%), followed by those who graduated from a non-medical university (18.6%) or from a vocational school of higher education (16.9%). Moreover, 84% of the respondents completed postgraduate training (specialisation in a specific field of nursing).

The mean work experience was 18.9 years (SD = 11.86 years). The majority of the respondents were employed under a contract of employment (84.7%), others worked in a two-shift system (61.2%) and worked in a clinical hospital (25.7%), in a voivodship (22.7%), county (20.7%), or a municipal hospital (13%). The other respondents worked in primary healthcare facilities (16.9%), curative care centres (3.5%), or nursing homes (1.3%). Further analysis of the employment profile revealed data on the place of work: a medical treatment ward (33.3%), noninvasive treatment ward (19.9%), paediatric ward (15.5%), or intensive care unit and emergency unit (total: 14.3%); work at a different ward was declared by 0.5% of the respondents, and 16.5% of them did not choose any of the options (Appendix A).

More than half of the surveyed nurses (64.5%) did not use EBNP in their work, 29.6% answered affirmatively, while 5.9% did not give an answer. In total, 61.9% of the respondents declared that they learned about the subject of EBNP in the course of their education, 34.1% did not have knowledge of this issue, and 4% of the respondents did not provide any answers. The majority of nurses, 40.5% learned about EBNP during their studies, 14% during postgraduate education, and 10.7% during self-education. When examining the use of EBNP in the work of the respondents, the following data were obtained: half of the respondents 418 (50.4%) did not use it, 356 (42.9%) used it, and 56 people (6.7%) did not provide an answer (data not shown).

### 3.2. The Level of Knowledge of, Attitudes toward, and Skills in EBNP According to the EBPPQ

An EBPPQ questionnaire was used to evaluate the knowledge of, attitudes toward, and skills in EBNP. According to the results, the highest scores were given to “EBNP-related skills” (M = 5.56 ± 1.84), followed by “Level of knowledge of scientific research terminology” (M = 5.50 ± 2.03) and “Other aspects of EBNP” (M = 5.50 ± 1.80). The other subscales included: “Attitude towards expanding one’s competence in EBNP”, “Attitude towards selected aspects of EBNP at work”, and “Frequency of using various EBNP components in everyday clinical work” and were estimated at similar levels (M from 5.48 to 5.49, SD from 1.84 to 1.91) (Table 2).

### 3.3. Impact of Sociodemographic Variables on Knowledge of, Attitudes toward, and Skills in EBNP among Nurses According to the EBPPQ

This study analysed the impact of sociodemographic variables (age, marital status, education, and work experience) on the knowledge of, attitudes toward, and skills in EBNP according to the EBPPQ.

The results revealed a statistically significant negative correlation (*p* < 0.05) between age and the subscales “Frequency of using various EBNP components in everyday clinical work” (r = −0.13), “Level of knowledge of scientific research terminology” (r = −0.07), and “EBNP-related skills” (r = −0.12) (Table 3).

Three groups of nurses, single, in a formal relationship, and those in an informal relationship, were analysed in terms of the marital status impact on the knowledge of, attitudes toward, and skills in EBNP according to the EBPPQ. A statistically significant difference in the mean level of “EBNP-related skills” was demonstrated in the study (F_(2,811)_ = 3.644; *p* = 0.027; η^2^ = 0.009; 95% CI [0.001; 0.024]). The mean level of these skills was found to be significantly higher in individuals in informal relationships than in those in formal relationships (M: 5.88 vs. 5.44; Scheffe test: *p* = 0.032). However, no significant differences were found between the group of single individuals and those in formal or informal relationships (Table 4).

The impact of the respondents’ educational background on their knowledge of, attitudes toward, and skills in EBNP was considered in two groups: respondents with secondary (medical secondary or postsecondary school) or university education (bachelor or master of nursing studies).

The results revealed a significant difference (*p* < 0.05) between the mean level of “The attitude towards expanding one’s competence in EBNP”, “Level of knowledge of scientific research terminology”, “Frequency of using various EBNP components in everyday clinical work”, “EBNP-related skills”, and “Other aspects of EBNP” in the groups under study.

A higher mean level of a subscale under study was observed in all of the statistically significant correlations in a group of respondents with university education compared with those who completed secondary or postsecondary school. The strongest impact of the educational background on the attributes under study was observed for the “EBNP-related skills” subscale (d = 0.46; 95% CI [0.26; 0.63]), (Table 5).

### 3.4. Impact of Work-Related Variables on Knowledge of, Attitudes toward, and Skills in EBNP among Nurses According to the EBPPQ

A weak negative correlation was revealed in terms of the work experience impact on the knowledge of, attitudes toward, and skills in EBNP according to the EBPPQ, concerning: “Frequency of using various EBNP components in everyday clinical work” (r = −0.13; *p* < 0.001) and “EBNP-related skills” (r = −0.10; *p* = 0.004) (Table 6).

This study analysed the impact of specialisation training on the knowledge of, attitudes toward, and skills in EBNP according to the EBPPQ. The results revealed statistically significant differences (*p* < 0.05) in the knowledge of, attitudes toward, and skills in EBNP among nurses according to the EBPPQ as affected by specialisation training in a specific field of nursing. A higher level of the subscale was demonstrated in the group of respondents with a specialisation. The strongest impact of specialisation on the subscales was observed for “The attitude to expanding one’s competence in EBNP” (M: 5.74 vs. 5.37; *p* = 0.007; 95% Cl [0.05; 0.35]) (Table 7).

### 3.5. Impact of Work-Related Variables on the Knowledge of, Attitudes toward, and Skills in EBNP among Nurses According to the EBPPQ and Regression Analysis

We used regression analysis, introducing variables to the full multivariate model and the stepwise model in order to generate the best model explaining the highest possible level of the variance of the dependent variable.

#### 3.5.1. Regression Analysis for the Subscale ‘Attitude towards Expanding One’s Competence in EBNP’

The positive attitude toward expanding one’s own EBNP competencies was shown to increase with work experience (β = 0.28, *p* = 0.009). Those who completed training organized by the Board of Nursing had significantly worse attitudes toward expanding their EBNP competencies than those who completed training organized by a university (β = −0.11, *p* = 0.018). People who worked in the Primary Nursing system had a significantly better attitude towards expanding their EBNP competencies than those working in other systems (β = 0.17, *p* = 0.028). People who had the subject of EBNP in the course of their education had a significantly better attitude towards expanding their competencies regarding EBNP than individuals without such experience during their training (β = 0.31, *p* < 0.001). People using EBNP in their work had a significantly better attitude towards expanding their EBNP competencies than individuals without such opportunities in the workplace (β = 0.34, *p* < 0.001) (Table 8). The presented model was statistically significant (F (25, 366) = 8.707, *p* < 0.001). The independent variables together explained approximately 33% of the variation in the dependent variable (adjusted-R^2^ = 0.33, SEE = 1.51).

The stepwise model additionally confirmed that the negative attitude toward expanding one’s own EBNP competencies increased with age (β = −0.20, *p* = 0.046), and individuals who worked in a hospital had significantly worse attitudes towards expanding their EBNP competencies than those who worked in other workplaces (β = −0.11, *p* = 0.011) (Table 9). The presented model was statistically significant (F (7, 384) = 28.079, *p* < 0.001). The independent variables together explained approximately 33% of the variation in the dependent variable (adjusted-R^2^ = 0.33, SEE = 1.52).

#### 3.5.2. Regression Analysis for the Subscale ‘Attitude towards Selected Aspects of EBNP at Work’

People who worked in clinical hospitals (β = 0.19, *p* = 0.038), district hospitals (β = 0.17, *p* = 0.031), and primary care centres (β = 0.15, *p* = 0.018) had significantly better attitudes towards selected aspects of EBNP at work than people who worked in other workplaces. Moreover, people who worked in surgical wards had significantly better attitudes towards selected aspects of EBNP at work than those who worked in other wards (β = 0.21, *p* = 0.045). People who came into contact with the topic of EBNP in the workplace had significantly better attitudes towards using selected aspects of EBNP at work than people without such experience (β = 0.16, *p* < 0.010) (Table 10). The presented model was statistically significant (F (25, 366) = 1.777, *p* = 0.013). The independent variables together explained approximately 5% of the variation in the dependent variable (adjusted-R^2^ = 0.05, SEE = 1.90).

In the multivariate model with backward elimination, it was additionally confirmed that people who worked in hospitals had significantly better attitudes towards selected aspects of EBNP at work than those in other workplaces (β = 0.12, *p* = 0.033) (Table 11). The presented model was statistically significant (F (25, 366) = 4.192, *p* = 0.001). The independent variables together explained approximately 4% of the variation in the dependent variable (adjusted-R^2^ = 0.04, SEE = 1.90).

#### 3.5.3. Regression Analysis for the Subscale ‘Level of Knowledge of Scientific Research Terminology’

People who came into contact with the topic of EBNP in the workplace had a significantly better knowledge of the terminology related to scientific research than those without such experience (β = 0.18, *p* = 0.001) (Appendix A). The presented model was statistically significant (F (5, 386) = 3.334, *p* < 0.001). The independent variables together explained approximately 13% of the variation in the dependent variable (adjusted-R^2^ = 0.13, SEE = 1.93).

It was confirmed in the stepwise model that people working under a hospital employment contract had a significantly better level of knowledge of terminology related to scientific research than those working under a civil contract (β = −0.10, *p* = 0.034) (Appendix A). The presented model was statistically significant (F (3, 386) = 21.919, *p* < 0.001). The independent variables together explained approximately 14% of the variation in the dependent variable (adjusted-R^2^ = 0.14, SEE = 1.92).

#### 3.5.4. Regression Analysis for the Subscale ‘Frequency of Using Various EBNP Components in Everyday Clinical Work’

People who used EBNP in their work were significantly more likely to apply individual elements of EBNP in their daily clinical work than those without such opportunities in the workplace (β = 0.17, *p* = 0.009). The presented model was statistically significant (F (25, 366) = 1.707, *p* = 0.020). The independent variables together explained approximately 4% of the variation in the dependent variable (adjusted-R^2^ = 0.04, SEE = 1.896) (Appendix A).

It was noted in the stepwise model that the older the nurses, the less frequent the use of individual EBNP elements in their daily clinical work (β = −0.17, *p* = 0.001), and people who used EBNP in their work used individual elements of EBNP in everyday clinical work definitely more often than those without such opportunities in the workplace (β = 0.20, *p* < 0.001) (Appendix A). The presented model was statistically significant (F (2, 389) = 14.015, *p* < 0.001). The independent variables together explained approximately 6% of the variation in the dependent variable (adjusted-R^2^ = 0.062, SEE = 1.877).

#### 3.5.5. Regression Analysis for the Subscale ‘EBNP-Related Skills’

People who completed training organized by the Board of Nursing had significantly worse EBNP-related skills than those who completed training organized by a university (β = −0.14, *p* = 0.010). Similarly, people employed in several places had worse EBNP-related skills than those employed only in one workplace (β = −0.14, *p* = 0.013). People who used individual elements of EBNP in everyday clinical work had significantly better EBNP-related skills than those without such opportunities in the workplace (β = 0.17, *p* = 0.009). The presented model was statistically significant (F (25, 366) = 2.076, *p* = 0.002). The independent variables together explained approximately 6% of the variation in the dependent variable (adjusted-R^2^ = 0.06, SEE = 1.697) (Appendix A).

It was observed in the stepwise model that people who completed training organized by the Board of Nursing had significantly worse EBNP-related skills than those who had completed training organized by a university (β = −0.12, *p* = 0.018). Similarly, people employed in several places had worse EBNP-related skills than those employed only in one workplace (β = −0.10, *p* = 0.040). People who used EBNP in their work had better EBNP-related skills (β = 0.24, *p* < 0.001) than those without such opportunities in the workplace (Appendix A). The presented model was statistically significant (F (4, 385) = 8.812, *p* < 0.001). The independent variables together explained approximately 7% of the variation in the dependent variable (adjusted-R^2^ = 0.074, SEE = 1.688).

## 4. Discussion

### 4.1. Impact of Sociodemographic Variables on the Knowledge of, Attitudes toward, and Skills in EBNP among Nurses According to the EBPPQ

A contemporary healthcare system requires nurses to be properly educated and highly qualified. The aim is to improve the operation of healthcare facilities by setting professional challenges, including the obligation to conduct scientific research aimed at developing the best patient care methods. Despite many benefits brought by EBNP, this concept is not systematically implemented in everyday practice in Poland [15].

This study examined sociodemographic and work-related variables to determine their impact on the knowledge of, attitudes toward, and skills in EBNP among nurses according to the Evidence-Based Practice Profile Questionnaire (EBPPQ).

The analysis performed for this study concerning the impact of age revealed a statistically significant negative correlation. Moreover, regarding marital status, the study demonstrated a statistically significant difference in the mean level of EBNP-related skills. The mean level of these skills was found to be significantly higher in individuals in informal relationships. Similar trends concerning the impact of age, educational background, and the type of higher education institution from which nurses graduated were observed among the midwives recruited for the study at the beginning of the 2014 spring session of state specialisation exams organised by the Post-Graduate Training Centre for Nurses and Midwives [16]. In their study, Wąsowska and Kózka confirmed that both younger nurses (approximately 36 years old) and those with higher education (bachelor’s or master’s degree) exhibited better knowledge of EBNP-related issues [15]. There are many papers in the literature describing the attitudes of various nurses’ groups to EBNP. However, few of them take into account the diversity of the nurses’ educational backgrounds [17].

The current study analysed the level of the nurses’ knowledge of, attitudes toward, and skills in terms of EBNP implementation at work. It was found that their studies had a significant impact on expanding their EBNP knowledge. An analysis of data from this study on the education background impact on the knowledge of, attitudes toward, and skills in EBNP revealed a statistically significant difference in the mean level of attitudes towards expanding their EBNP-related skills and in the mean level of the knowledge of scientific research-related terminology. A statistically significant correlation with respect to EBNP-related skills was demonstrated in the group of respondents with university education compared to those who completed secondary or postsecondary school. This was also confirmed by the findings of the study conducted by Eizenberg in 2007 in a group of 243 nurses in Israel. It showed that holding a bachelor’s degree made the attitudes toward clinical trials and the application of reliable scientific research in everyday work more positive than in the group of nurses with secondary education [18]. A 2014 study presented by the Faculty of Health Sciences at the Medical University of Warsaw contained similar findings in a group of epidemiological nurses taking a specialisation exam in epidemiology. It was shown that most nurses would apply scientific evidence in their work more frequently [19]. A team of researchers from the Medical University of Warsaw analysed the knowledge of, attitudes toward, and skills in EBNP in everyday work depending on the educational background of a group of 820 nurses. The findings were similar to the current study and are indicative of significant differences between the groups. These differences concern the attitude of individuals with a university education compared with individuals with secondary education or those holding a bachelor’s degree. Most of the respondents (over 60%) thought they should use scientific evidence in their work more frequently, and 57% of them found scientific reports useful. The authors demonstrated significant differences between the group of respondents with university education, nurses with secondary medical education, and those holding a bachelor’s degree [20].

### 4.2. Impact of Work-Related Variables on the Knowledge of, Attitudes toward, and Skills in EBNP among Nurses According to the EBPPQ

The area and the level of complexity of nurses’ knowledge of, attitudes toward, and skills in the constant development of medical sciences and technology, indicate the directions in which specialist knowledge is developing, regardless of one’s educational background or work experience. The essence of the professional perception of a patient’s clinical situation lies in the regular updating of knowledge and permanent learning.

An analysis performed for this study with respect to the impact of work experience on the nurses’ knowledge of, attitudes toward, and skills in EBNP according to the EBPPQ revealed a weak negative correlation in terms of the frequency of using various EBNP components in everyday clinical work and EBNP-related skills. Further data analysis showed statistically significant differences in the nurses’ knowledge of, attitudes toward, and skills in EBNP according to the EBPPQ depending on the nursing specialisation. A higher mean attribute level was demonstrated in the group of respondents with a specialisation. The strongest impact was observed for the attitude to expanding one’s competence in EBNP. Separate findings were presented by researchers at the Medical University of Warsaw, who analysed the correlation between the level of knowledge and types of attitudes depending on the work experience. The findings for the domain “use of EBP in nurses’ professional practice” are indicative of much more extensive knowledge and a positive attitude of respondents with little work experience towards the use of scientific literature and improvement in practice quality owing to EBP than in nurses with longer work experience. The longer the nurses’ work experience, the greater the workload and a critical attitude to new ideas in performing their nursing duties [21]. After two years, a team of researchers at the same university performed a cross-sectional validation study to evaluate the level of the knowledge, skills, and attitudes of Polish nurses with respect to the EBNP in correlation with their educational background and work experience. A weak negative correlation was revealed in terms of the frequency of using various EBNP component in everyday clinical work and EBNP-related skills. The data analysis showed statistically significant differences in the nurses’ knowledge of, attitudes toward, and skills in EBNP according to the EBPPQ depending on the nursing specialisation. A higher level of the attribute was demonstrated in the group of respondents with a specialisation. The strongest impact was observed for the attitude to expanding one’s competence in EBNP.

According to one of the study conclusions, nurses must continue to expand their knowledge in EBNP by participation in various forms of postgraduate training and by seeking new information in order to create a powerful and positive EBNP profile in nursing [22].

A study conducted in Southeast Asia in 2003 of 600 nurses in four Malaysian hospitals showed a significant difference in their commitment to nursing research between groups with long and short work experience. Nurses with a short work experience thought that EBNP increased their workload, because it obligated them to read the latest medical and scientific reports [23]. The level of knowledge of, attitude toward, and skills in implementing evidence-based practices was evaluated in a group of 200 Jordanian nurses in intensive care units. The personnel participating in the study were shown to possess EBNP-related knowledge and to have a positive attitude to evidence-based practices, and higher personnel qualifications and training in EBNP contributed to its practical implementation [24].

### 4.3. Regression Analysis for the Subscale

Our study showed that there were several significant predictor variables that influenced the subscales of the Evidence-Based Practice Profile Questionnaire. A positive attitude towards expanding one’s own EBNP competencies was found to increase with work experience. Moreover, work in the Primary Nursing system, completion of training courses organized by a university, contact with EBNP-related issues during education, and using EBNP at work had a positive impact on expanding the nurses’ competencies.

Aynalem et al. [25] assert that there are seven variables statistically associated with the use of EBP after adjusting for confounding factors. These are marital status, work experience, knowledge of EBP, communication skills, training on EBP, availability of evidence-based guidelines, and access to the internet. The aforementioned research found that unmarried nurses were better at using EBP than their married counterparts. These findings contradict the results obtained by Barako et al. [26], who claimed that marital status was not related to the use of the EBP. Furthermore, Aynalem et al. found that work experience was significantly related to the use of EBP. They observed that those who had been in the profession for more than 10 years were less likely to use EBP compared to those who had been in the profession for a shorter time. These results are consistent with those reported by Heydari et al. [27]. In contrast, Hadgu et al. [28] and Dalheim et al. [29] noted that work experience was not related to EBP.

Numerous studies have also shown that good knowledge of EBP [28,30,31,32] or the availability of evidence-based guidelines [30,33] positively influence the use of EBP in professional work.

Ahmad Ghaus et al. [34], on the other hand, found that sex, race, average number of patients observed during the day, and neutral attitudes were significantly related to using EBM in the workplace. This is consistent with the report by Zanaridah et al. [35], who noticed that work experience was also significantly related to the EBM practice. According to Abdel-Kareem A. et al. [36], sex, specialisation, medical qualifications, and previous training in EBM were factors that influenced attitudes towards EBM practice.

As numerous studies show, the accessibility to databases/online resources in the workplace has a positive effect on the EBM practice [37,38]. Lafuente-Lafuente C. et al. [37] emphasized that the lack of access to information is a major barrier to the practice of EBM. Therefore, it is essential that the EBM libraries are widely available, easy to use, and have electronic databases.

Yoo et al. [39] proved that the EBNP knowledge model (β = 0.15) and organizational readiness (β = 0.36) were significant predictors of EBNP implementation; the model predicted 22.2% of variance of the EBP implementation (F = 10.098, *p* < 0.001). Moreover, age was strongly correlated with clinical experience and was excluded from the independent variables. Furthermore, after analysing the differences in the main variables by nurse characteristics, Yoo et al. reported that a higher educational status and experience in conducting or participating in research had a significant impact on the implementation of EBP. These results are in line with other findings which confirm that the more research activities, the higher the level of knowledge, beliefs, and organizational readiness of EBP, and the higher the level of EBNP implementation [40,41,42,43]. It is therefore essential to ensure that nurses are directly involved in the planning and implementation of research projects related to EBNP.

### 4.4. Applying the Results of the Study into the Curriculum of Nursing Studies

The results of this study should be carefully incorporated into the curriculum of nursing studies. The curriculum should include the aspects of the role of nursing faculty in nursing education research and activities, as well as the role of reviews the literature related to nursing education and practice in the context of further studies and practice. The curriculum should include: evidence-based nursing (EBM and EBNP), components of the EBNP process, ethical principles in introducing and the propagation of research in nursing, and the use of EBM-based good practices for the development of the profession [13]. Professional nursing care should be based on scientific models of nursing, supported by evidence, based on the patient’s needs, problems, life situation, as well as expectations in health and illness. Therefore, EBN should play a significant role in nurses’ work.

## 5. Conclusions

Sociodemographic variables with an impact on the knowledge of, attitudes toward, and skills in EBNP include age, marital status, and educational background. Older nurses found it problematic to apply EBNP, and their level of relevant knowledge and skills was lower; whereas those with a university education possessed the best EBNP-related skills, and they were also more eager to expand them and apply them in their work.The work-related variables with an impact on the knowledge of, attitudes toward, and skills in EBNP among nurses included work experience, which had a negative effect on applying EBNP and the skills associated with it. The type of school that the nurses graduated from was another variable—nearly all the subscales showed better achievements among medical university graduates. Similarly, nursing specialisation training had a great impact on expanding nurses’ competence in EBNP.In order to introduce EBNP into the daily practice of nurses in Poland, it is essential to minimize barriers, strive to build the necessary infrastructure, and develop a curriculum that will not only support but also assess each nurse’s knowledge of EBNP.There were many variables statistically related to the use of EBNP in the daily work of Polish nurses. The main predictive variables influencing the subscales of the Evidence-Based Practice Profile Questionnaire were marital status, work experience, knowledge of EBNP, the EBNP training site, and the availability of evidence-based guidelines.

## 6. Limitation

Considering the nurse training system in Poland, it should be emphasised that the study findings concerning the sociodemographic data with an impact on the knowledge of, attitudes toward, and skills in EBNP may have been affected by the respondents’ age and—closely related to it—their educational background, which depends on the under- and postgraduate training system in Poland. The study findings with respect to work-related variables with an impact on the knowledge of, attitudes toward, and skills in EBNP may have been affected by diverse work organisation systems, organizational culture, regional characteristics within a healthcare facility, as well as the composition of nursing staff. We will improve these variable factors in our future studies.

Another limitation of our research may be the random selection of the study sample. Further, our research was conducted in only some parts of Poland. Hence, the results of this study may not be representative of all nurses and cannot be generalized. However, the results of this study should be carefully incorporated into the curriculum of nursing studies.

## Figures and Tables

**Table 1 ijerph-19-08548-t001:** Characterisation of the group under study.

Variable		*n*	%
**Sex**	Female	790	95.2
	Male	33	4
	No data available	7	0.8
**Age group [years]**	≤30	186	22.4
	31–40	121	14.6
	41–50	348	41.9
	>50	153	18.4
	No data available	22	2.7
**Place of residence**	Village	170	20.5
	Town with a population under 10 thousand	84	10.1
	Town with a population under 100 thousand	211	25.4
	Town with a population over 100 thousand	294	35.4
	City—voivodship capital	62	7.5
	No data available	9	1.1
**Marital status**	Single	131	15.8
	Informal relationship	156	18.8
	Formal relationship	534	64.3
	No data available	9	1.1
**Education**	Medical secondary school	121	14.6
	Medical postsecondary school	65	7.8
	Bachelor of nursing studies	364	43.9
	Master of nursing studies	267	32.2
	Doctorate	2	0.2
	No data available	11	1.3

**Table 2 ijerph-19-08548-t002:** Knowledge of, attitudes toward, and skills in EBNP according to the EBPPQ (sten scale) among the nurses under study.

Subscale [sten]	M	SD	CV [%]
Attitude towards expanding one’s competence in EBNP	5.49	1.87	34.01
Attitude towards selected aspects of EBNP at work	5.48	1.91	34.92
Level of knowledge of scientific research terminology	5.50	2.03	36.87
Frequency of using various EBNP components in everyday clinical work	5.48	1.90	34.68
EBNP-related skills	5.56	1.84	33.12
Other aspects of EBNP	5.50	1.80	32.79

M—mean, SD—standard deviation, CV—coefficient of variation.

**Table 3 ijerph-19-08548-t003:** Impact of the respondents’ age on the knowledge of, attitudes toward, and skills in EBNP according to the EBPPQ.

Subscales	r-Pearson	t	*p*
Attitude towards expanding one’s competence in EBNP	0.01	0.241	0.810
Attitude towards selected aspects of EBNP at work	−0.01	−0.234	0.815
Level of knowledge of scientific research terminology	−0.07	−2.016	0.044
Frequency of using various EBNP components in everyday clinical work	−0.13	−3.620	0.000
EBNP-related skills	−0.12	−3.510	0.000
Other aspects of EBNP	−0.03	−0.841	0.400

r-Pearson—correlation coefficient. t—the value of the statistic for testing the significance of the correlation coefficient. *p*—test probability.

**Table 4 ijerph-19-08548-t004:** Impact of the respondents’ marital status on the knowledge of, attitudes toward, and skills in EBNP according to the EBPPQ.

Subscale	Single Individual		Formal Relationship		Informal Relationship		F	*p* *
	M	SD	M	SD	M	SD		
Attitude towards expanding one’s competence in EBNP	5.46	1.88	5.49	1.89	5.55	1.76	0.096	0.908
Attitude towards selected aspects of EBNP at work	5.51	1.80	5.41	1.97	5.66	1.76	1.074	0.342
Level of knowledge of scientific research terminology	5.68	1.95	5.42	2.04	5.65	2.06	1.330	0.265
Frequency of using various EBNP components in everyday clinical work	5.55	1.91	5.38	1.91	5.71	1.84	1.867	0.155
EBNP-related skills	5.64	1.89	5.44	1.87	5.88	1.66	3.644	0.027
Other aspects of EBNP	5.48	1.75	5.46	1.81	5.67	1.84	0.841	0.432

M—mean, SD—standard deviation, *p*—test probability, * ANOVA—one-way.

**Table 5 ijerph-19-08548-t005:** Impact of the respondents’ educational background on the knowledge of, attitudes toward, and skills in EBNP according to the EBPPQ.

Subscale	Secondary		University		t_(df = 802)_	* *p*	d	95% CI
	M	SD	M	SD				
Attitude towards expanding one’s competence in EBNP	5.03	1.92	5.61	1.83	−3.738	0.000	0.31	0.15; 0.48
Attitude towards selected aspects of EBNP at work	5.52	2.05	5.47	1.83	0.331	0.740	–	–
Level of knowledge of scientific research terminology	4.99	2.01	5.62	2.00	−3.698	0.000	0.32	0.15; 0.48
Frequency of using various EBNP components in everyday clinical work	4.95	1.93	5.58	1.83	−4.031	0.000	0.34	0.17; 0.51
EBNP-related skills	4.91	1.92	5.73	1.75	−5.430	0.000	0.46	0.29; 0.63
Other aspects of EBNP	5.19	1.75	5.57	1.79	−2.499	0.013	0.21	0.05; 0.38

M—mean, SD—standard deviation, d—Cohen’s coefficient, CI—confidence interval, *p* *—level of significance, Student *t*-test.

**Table 6 ijerph-19-08548-t006:** Impact of the respondents’ work experience on the knowledge of, attitudes toward, and skills in EBNP according to the EBPPQ.

Subscales	r-Pearson	t	*p*
Attitude towards expanding one’s competence in EBNP	0.03	0.923	0.356
Attitude towards selected aspects of EBNP at work	0.01	0.301	0.763
Level of knowledge of scientific research terminology	−0.06	−1.547	0.122
Frequency of using various EBNP components in everyday clinical work	−0.13	−3.731	0.000
EBNP-related skills	−0.10	−2.903	0.004
Other aspects of EBNP	−0.01	−0.236	0.813

r-Pearson—correlation coefficient. t—the value of the statistic for testing significance of correlation coefficient, *p*—test probability.

**Table 7 ijerph-19-08548-t007:** Impact of the respondents’ nursing specialisation on the knowledge of, attitudes toward, and skills in EBNP according to the EBPPQ.

Subscale	Specialisation Training				t_(df = 813)_	*p* *	d	95% CI
	No		Yes					
	M	SD	M	SD				
Attitude towards expanding one’s competence in EBNP	5.37	1.85	5.74	1.87	−2.717	0.007	0.20	0.05; 0.35
Attitude towards selected aspects of EBNP at work	5.56	1.79	5.30	2.06	1.878	0.061	–	–
Level of knowledge of scientific research terminology	5.45	1.95	5.63	2.19	−1.185	0.237	–	–
Frequency of using various EBNP components in everyday clinical work	5.43	1.87	5.54	1.94	−0.777	0.438	–	–
EBNP-related skills	5.54	1.87	5.64	1.75	−0.743	0.457	–	–
Other aspects of EBNP	5.44	1.76	5.62	1.86	−1.407	0.160	–	–

M—mean, SD—standard deviation, d—Cohen’s coefficient, CI—confidence interval, *p* *—level of significance, Student *t* test.

**Table 8 ijerph-19-08548-t008:** Regression analysis for the full multivariate model for the subscale ‘Attitude towards expanding one’s competence in EBNP’.

Variable	Level	β	+95% CI	−95% CI	t	*p*-Value
Intercept					6.975	<0.001
Age		−0.20	−0.41	0.01	−1.840	0.067
Work experience		0.28	0.07	0.49	2.620	0.009
Trainings/courses	No (ref.)					
	Yes	0.02	−0.07	0.11	0.453	0.650
Specialisation	No (ref.)	0.06	−0.03	0.15	1.209	0.227
	Yes					
Trainings organized by:	University (ref.)					
	Private organization	0.02	−0.07	0.11	0.426	0.671
	Board of Nursing	−0.11	−0.19	−0.02	−2.377	0.018
Employment relationship	Civil contract (ref.)					
	Employment contract	−0.04	−0.13	0.04	−0.969	0.333
Number of workplaces	1 (ref.)					
	>1	−0.06	−0.15	0.04	−1.207	0.228
Workplace	Clinical hospital	−0.04	−0.19	0.11	−0.535	0.593
	Provincial hospital	0.07	−0.07	0.21	0.968	0.334
	City hospital	0.09	−0.03	0.20	1.455	0.146
	District hospital	0.01	−0.12	0.15	0.208	0.835
	Residential home	−0.01	−0.10	0.08	−0.286	0.775
	Primary Care Center	0.04	−0.07	0.14	0.698	0.485
	Residential medical care facility	0.01	−0.07	0.10	0.268	0.788
Hospital ward	Other (ref.)					
	Paediatric	−0.15	−0.31	0.00	−1.937	0.054
	Surgical	−0.11	−0.29	0.06	−1.260	0.208
	Noninvasive treatment	−0.13	−0.29	0.03	−1.652	0.099
	Intensive Care Unit (ICU), Emergency Department	−0.03	−0.18	0.11	−0.456	0.649
Work organization system	Other (ref.)					
	Functional Model of Care	−0.02	−0.14	0.11	−0.246	0.806
	Small-Team Model	−0.12	−0.24	0.01	−1.859	0.064
	Primary Nursing	0.17	0.02	0.33	2.208	0.028
Is EBNP used in the workplace?	No (ref.)					
	Yes	−0.06	−0.16	0.05	−1.062	0.289
Did they have the subject of EBNP in the course of education?	No (ref.)					
	Yes	0.31	0.21	0.40	6.482	<0.001
Do they use EBNP in their work?	No (ref.)					
	Yes	0.34	0.23	0.44	6.141	<0.001

**Table 9 ijerph-19-08548-t009:** Regression analysis for the stepwise model for the subscale ‘Attitude towards expanding one’s competence in EBNP’.

Variable	Level	β	+95% CI	−95% CI	t	*p*-Value
Intercept					9.950	<0.001
Did they have the subject of EBNP in the course of education?	No (ref.)					
	Yes	0.32	0.23	0.41	6.837	<0.001
Trainings organized by:	University (ref.)					
	Private organization	0.01	−0.08	0.09	0.178	0.859
	Board of Nursing	−0.11	−0.19	−0.02	−2.483	0.013
Do they use EBNP in their work?	No (ref.)					
	Yes	0.30	0.21	0.39	6.420	<0.001
Workplace: clinical hospital	No (ref.)					
	Yes	−0.11	−0.19	−0.03	−2.571	0.011
Work experience		0.27	0.07	0.47	2.640	0.009
Age		−0.20	−0.40	−0.00	−2.004	0.046

**Table 10 ijerph-19-08548-t010:** Regression analysis for the full multivariate model for the subscale ‘Attitude towards selected aspects of EBNP at work’.

Variable	Level	β	+95% CI	−95% CI	t	*p*-Value
Intercept					5.129	<0.001
Age		0.01	−0.24	0.26	0.082	0.935
Work experience		−0.04	−0.29	0.21	−0.312	0.755
Trainings/courses	No (ref.)					
	Yes	−0.04	−0.14	0.07	−0.690	0.491
Specialisation	No (ref.)					
	Yes	−0.11	−0.22	0.00	−1.953	0.052
Trainings organized by:	University (ref.)					
	Private organization	0.02	−0.08	0.13	0.453	0.651
	Board of Nursing	−0.04	−0.14	0.07	−0.732	0.465
Employment relationship	Civil contract (ref.)					
	Employment contract	−0.08	−0.19	0.02	−1.595	0.111
Number of workplaces	1 (ref.)					
	>1	0.04	−0.06	0.15	0.800	0.424
Workplace	Clinical hospital	0.19	0.01	0.36	2.079	0.038
	Provincial hospital	0.08	−0.09	0.25	0.906	0.365
	City hospital	0.05	−0.09	0.19	0.673	0.502
	District hospital	0.17	0.02	0.33	2.171	0.031
	Residential home	0.09	−0.02	0.19	1.658	0.098
	Primary care centers	0.15	0.03	0.27	2.367	0.018
	Residential medical care facility	−0.03	−0.13	0.07	−0.564	0.573
Hospital ward	Other (ref.)					
	Paediatric	0.09	−0.09	0.28	0.998	0.319
	Surgical	0.21	0.00	0.42	2.010	0.045
	Noninvasive treatment	0.19	−0.00	0.38	1.950	0.052
	Intensive Care Unit (ICU), Emergency Department	0.12	−0.06	0.29	1.326	0.186
Work organization system	Other (ref.)					
	Functional Model of Care	0.11	−0.04	0.26	1.418	0.157
	Small-Team Model	0.00	−0.14	0.15	0.052	0.959
	Primary Nursing	−0.01	−0.20	0.17	−0.150	0.881
Is EBNP used in the workplace?	No (ref.)					
	Yes	0.16	0.04	0.29	2.588	0.010
Did they have the subject of EBNP in the course of education?	No (ref.)					
	Yes	−0.02	−0.14	0.09	−0.421	0.674
Do they use EBNP in their work?	No (ref.)					
	Yes	−0.04	−0.17	0.09	−0.591	0.555

**Table 11 ijerph-19-08548-t011:** Regression analysis for the stepwise model for the subscale ‘Attitude towards selected aspects of EBNP at work’.

Variable	Level	β	+95% CI	−95% CI	t	*p*-Value
Intercept					31.035	<0.001
Workplace: district hospital	No (ref.)					
	Yes	0.12	0.02	0.23	2.306	0.022
Specialisation	No (ref.)					
	Yes	−0.12	−0.22	−0.02	−2.368	0.018
Workplace: primary care center	No (ref.)					
	Yes	0.11	0.01	0.21	2.213	0.027
Is EBNP used in the workplace?	No (ref.)					
	Yes	0.12	0.02	0.22	2.446	0.015
Workplace: clinical hospital	No (ref.)					
	Yes	0.12	0.01	0.22	2.136	0.033

## Data Availability

Not applicable.

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
