# Peer review of "Sociodemographic and Work-Related Variables Affecting Knowledge of, Attitudes toward, and Skills in EBNP of Nurses According to an EBPPQ"

_ijerph, 2022, doi:10.3390/ijerph19148548_

Round 1

Reviewer 1 Report

This study seeks sociodemographic and work-related variables with an im
pact on knowledge, attitudes and skills in Evidence-Based Nursing Practice (EBNP). The study included 830 nurses from four voivodships in Poland.

The authors should provide more details on the method distributions the questionaires. Did the authors use paper or on line version? Please provide the respondent rate. How the authors recruited the respondents for this study.

The authors should expand the study limiations.

I suggest short some sentences from the Introduction section and Conclusion section

Author Response

Thank you for this suggestion, your comments helped us to improve the quality of the manuscript. 

Reviewer 2 Report

I have read with great interest the manuscript entitled “Sociodemographic and work-related variables affecting knowledge, attitudes and skills in EBNP of nurses according to an EBPPQ”. This study aimed to examine associations age, marital status, education, work experience, and nursing specialization with knowledges, attitudes, and skills in EBNP. A series of t-tests and analyses of variables were conducted. Some demographic and work-related variables were found to be associated with some aspects of EBNP. I think that the topic of this manuscript is important and is suitable for the nursing section of the journal. This is because dissemination of EBN represents one of unsolved issues in nursing and health care. However, multiples tests should increase the probability of at least one type 1 error, Statistical significance should be adjusted. It would be better to conduct a series of multiple regression analyses with demographic and work-related variables as the independent variables (predictors). Simultaneous examinations of these variables will prove which of these variables will be associated with outcomes. In addition, overall, the magnitudes of effects sizes were only small or small to medium. It seems necessary to refer to effect sizes and to interpret the results more critically. I recommend authors to reconsider how to analyze data and how to interpret results. Addressing my comments will change substantially the Results and Discussion sections. I have not listed up minor comments. I hope that my comments will be helpful.

Author Response

(The authors gave the same response as above.)

Reviewer 3 Report

Today, evidence-based medical services are being emphasized, it is very important to study evidence-based nursing practices. However, I think your research is not enough to be published in the journals Q1 and Q2. The reasons are as follows.

 First, it is academically meaningless to examine the difference in and correlation of EBNP with nothing special demographic profiles and educational or career variables. The results are understandable at the level of common sense.

Second, sample size is too big to interpret the results of correlational analysis and ANOVA or t-test. Of course, it can be interpreted in consideration of that. For example, if the correlational analysis shows that the r value is 0.13(p<0.001), the accountability is only 1.7%. If the r value is -0.07(p<0.05), it is only 0.5%. It's statistically significant, but it's practically meaningless. To be clinically and educationally interested, the accountability must be at least 10%.

Of course, even if the variable is an intrinsic personal trait and has low accountability, it can be interested academically in whether there is a moderator in the relationship between the two variables. But that's not the case with this kinds of general characteristics. And look at the means of variables that show significant differences in ANOVA or t-test. Can it be the difference interesting? You can see that the difference is not practically enough to be interested in.

Because of what I have described so far, I omit the comments related to writing the manuscript.

Author Response

(The authors gave the same response as above.)

Round 2

Reviewer 2 Report

Authors have addressed my concerns. 

Author Response

Dear Editors,

We take the liberty to thank you and the reviewers for insightful and careful evaluation of our article entitled “Sociodemographic and work-related variables affecting knowledge, attitudes and skills in EBNP of nurses according to an EBPPQ” by Katarzyna MÅ‚ynarska, Elżbieta Grochans, Magdalena Sylwia KamiÅ„ska (corresponding author), Anna Maria Cybulska, Mariusz Panczyk, Ewa Kupcewicz and for allowing us to resubmit a revised manuscript.

The comments helped us to improve the quality of the manuscript. We considered all comments and recommendations and responded to Reviewers’ questions. The correction throughout the manuscript were marked in yellow.

Our responses to the reviews are attached below.

Thank you for your consideration. We look forward to hearing from you.

Sincerely,

Magdalena Kamińska

Reviewer 3 Report

After the first round of reviewing, the researchers supplemented the manuscript a lot, and I evaluate that the quality has improved considerably as a research article. But there are somethings that I would like you to supplement in order to publish this manuscript. If those things are supplemented, I think that there is no big problem if the revised manuscript to be published in this journal. Those things are as follows.

1. There are a lot of tables that you suggested, but isn't there anything to omit because it's not salient finding?

2. The regression coefficient B is the non-standardized, and β means the standardized coefficient, so you just have to write the beta only, not βstd.

3. In discussion section, I would like you to suggest more specifically how to apply the results of this study for educating nursing students.

4. Because there are more limitations regarding to this study, you should add them more specifically.

Author Response

(The authors gave the same response as above.)
